# The Role of Gender in the Relationship Between Waist-to-Hip Ratio, Triglyceride–Glucose Index, and Insulin Resistance in Korean Children

**DOI:** 10.3390/healthcare13070823

**Published:** 2025-04-04

**Authors:** Seamon Kang, Xiaoming Qiu, Simon Kim, Hyunsik Kang

**Affiliations:** College of Sport Science, Sungkyunkwan University, Suwon 16419, Republic of Korea; seamon@skku.edu (S.K.); qq614972272@gmail.com (X.Q.); 90simon.kim@gmail.com (S.K.)

**Keywords:** sex, upper-body obesity, insulin resistance, children

## Abstract

**Background/Objectives:** Little is known about the relationship between obesity, the triglyceride–glucose (TyG) index, and insulin resistance (IR). This cross-sectional study of Korean children investigated whether the TyG index mediates the relationship between the waist-to-hip ratio (WHtR) and homeostatic model assessment for IR (HOMA-IR). **Methods:** Six-hundred-and-thirteen Korean children (320 boys and 293 girls) aged 9–12 years old participated in this study. The participants were classified as insulin-sensitive or insulin-resistant based on gender-specific cut-off values of HOMA-IR. The TyG index was calculated as follows: ln [fasting triglycerides (mg/dL) × fasting blood glucose (mg/dL)/2]. **Results:** Children with IR were older, more likely to be girls, and had fewer favorable metabolic risk factors than children without IR. A mediation analysis revealed that while WHtR has a direct effect on HOMA-IR, it also has an indirect effect on HOMA-IR through the TyG index. The bootstrapped 95% confidence interval (CI) confirmed that the TyG index had an indirect effect on the relationship between the WHtR and HOMA-IR (effect = 0.349, SE = 0.075, 95% CI [0.210, 0.504]). The interaction effect between the WHtR and sex for the TyG index was statistically significant (β = −1.369, SE = 0.631, 95% CI [−2.608, −0.129]), but it was no longer significant when vigorous physical activity was considered as a covariate. **Conclusions:** Our findings suggest that girls are more vulnerable than boys to an increase in the TyG index caused by an increase in WHtR. This gender disparity observed in the study needs to be investigated causally.

## 1. Introduction

Childhood obesity is a significant risk factor for insulin resistance (IR), which can lead to type-2 diabetes (T2D) and cardiovascular disease (CVD) later in life [1]. IR is a pathological condition in which the skeletal muscle, adipose tissue, and liver fail to respond normally to insulin [2]. IR is a hallmark of metabolic syndrome (MetS), which is characterized by central obesity, hyperglycemia, hypertension, and dyslipidemia, all of which lead to the development of T2D and CVD [3]. Unhealthy lifestyles, such as excessive caloric consumption, lack of physical activity, and sedentary behavior, have been blamed as the etiological risk factors underlying IR pathology [4].

Several biomarkers, such as fasting insulin, the homeostasis model assessment for IR (HOMA-IR), the quantitative insulin sensitivity check index, and an oral glucose tolerance test, have been developed and used to identify IR. Among these, HOMA-IR is the most effective and widely validated biomarker for determining glucose homeostasis within a healthy range [5]. A pathological link between obesity and this biomarker has been reported in pediatric populations [6,7]. Various cut-offs of HOMA-IR for whole-body IR have been developed for Asian children and adolescents, and their clinical utility in identifying youths at a high risk of whole-body IR has been proven in previous studies [8,9]. At the same time, previous research has also found sex differences in the association between the biomarker and obesity-related health conditions [10].

The triglyceride–glucose (TyG) index, a composite measure derived from fasting triglyceride and glucose levels, has been developed and used as an alternative for HOMA-IR [11]. The TyG index has been associated with obesity and related health conditions in adults from the USA [12] and China [13,14]. The TyG index is also positively associated with all-cause and diabetes-specific mortality in people with MetS [15]. The clinical value of the TyG index was critically reviewed and validated in a recent review and meta-analysis of 49,325 participants from 13 observational studies [16]. Compared with the gold standard hyperinsulinemic-euglycemic clamp for measuring IR, the TyG index is also easier to use, takes less time, and is more cost-effective for large-scale use [17]. However, there is little information about how sex influences the relationship between obesity, the TyG index, and whole-body IR in pediatric populations.

Sex-related differences in energy metabolism are well established [18], which may explain sex differences in IR [19,20]. Some studies have indicated that women have a lower TyG index than men, independent of body fat content [21,22], implying that women may be less insulin-resistant or more insulin-sensitive. A high TyG index has been linked to a high risk of subclinical atherosclerosis [23] and obstructive coronary artery disease in women without diabetes [24]. The association between the TyG index and T2D risk is stronger among women than among men [25]. A previous study also reports sex and age differences in the prevalence of obesity and related health conditions in children and adolescents in Korea [26]. Collectively, these findings indicate that sex may influence the relationship between obesity, the TyG index, and IR through several factors. This cross-sectional study investigated the influence of sex on the relationship between the waist-to-hip ratio (WHtR), TyG index, and HOMA-IR in a pediatric population using a moderated mediation model.

## 2. Materials and Methods

### 2.1. Study Participants

The study recruited participants using convenience sampling, which was based on geographical proximity and willingness to participate in the study. From June 2018 to March 2019, a total of 800 children (400 boys and 400 girls) aged 7–12 years and their parents from local elementary schools were invited to our orientation session where they received an oral presentation about the study.

As shown in Figure 1, 702 out of 800 students and their parents agreed to participate in the study (88% response rate). We then excluded those who did not provide fasting blood samples (10 boys and 30 girls), as well as those who did not take or complete body composition measurements (12 boys and 20 girls), or physical activity (PA) assessments (10 boys and 7 girls). The remaining 320 boys and 293 girls were included in the final data analysis (13% dropout rate). The sample size for testing the null hypothesis was calculated as 610 using the G*power 3.1.9.7 for Windows XP, with effect sizes of 0.197 (TyG index), 0.492 (WHtR), and 0.260 (HOMA-IR), an α error probability of 0.05, and a power (1-ß error probability) of 0.95. The appropriate sample size was 610. The Institutional Review Board of Human Study reviewed and approved this study (approval no SKKU 2018-06-005-003) according to the Declaration of Helsinki. Before participating in the study, all the participants and their parents provided written informed consent.

### 2.2. Measurement of Anthropometrics and Physical Activity

Body mass index (BMI) was calculated as weight (kg) divided by the square of height (m^2^). Waist circumference (WC) was measured at the midpoint between the lowest rib and iliac crest, and hip circumference was measured at the greatest part of the hip. WHtR was calculated as WC divided by the hip circumference.

Daily PA was monitored by using a uniaxial accelerometer (Kenz Lifecorder EX, Suzuken Co., Ltd., Nagoya, Japan). All the participants were asked to wear the device continuously, except while showering, for 7 consecutive days. At the end of the 7th day, the researchers manually shut down the device and transferred the recorded data to a personal computer. We classified the activity levels into nine categories (1.0–9.0) based on PA energy expenditure, which ranged between 1.8 and 8.3 estimated METs [27]. We further divided the nine PA levels into three categories, light PA (1.8–2.9 METs), moderate PA (3.6–5.2 METs), and vigorous PA (6.1–8.3 METs), using the software provided by the company.

### 2.3. Definition of Whole-Body Insulin Resistance

The fasting blood glucose (FBG), total cholesterol (TC), triglycerides (TG), and high-density lipoprotein cholesterol (HDLC) levels were measured using a Hitachi 7600 Automatic Analyzer 7600, (Hitachi, Tokyo, Japan). Systolic and diastolic blood pressures (BPs) were measured using a mercury sphygmomanometer. The HOMA-IR was computed using the previously available equation [28]. The participants were classified as insulin-sensitive or insulin-resistant based on the sex-specific cut-off values of HOMA-IR (3.54 and 3.69 for boys and girls, respectively) previously reported in Korean adolescents [8]. The TyG index was calculated using the following formula: TyG index = ln [fasting TG (mg/dL) × FBG (mg/dL)/2].

### 2.4. Statistics

Quantile–quantile plots were used to confirm the normality of the data distribution. All variables are expressed as means and standard deviations. Notably, Students’ *t*-tests and one-way ANOVAs with contrasts were performed to test for sex differences and linear trends in the measured parameters, respectively. The correlation coefficients between HOMA-IR and the measured parameters were calculated. As illustrated in Figure 2, we used Hayes’ process macro model 4 to conduct a mediation analysis, which involved testing for the relationship between the dependent variable Y (or HOMA-IR) and the independent variable X (or WHtR) via the proposed mediator M (or TyG index). We then tested whether sex influences the effect of X on M using the Hayes PROCESS macro model 7. Bias-corrected bootstrapping (n = 5000), and 95% confidence intervals (CIs) were used to evaluate the statistical significance of the mediation model. All statistical significance was tested at *p* = 0.05 using SPSS-PC (version 27.0; IBM Corporation, Armonk, NY, USA).

## 3. Results

Table 1 presents the descriptive statistics of the study participants. Children with IR were older, more likely to be girls, and had less favorable profiles of body fat parameters and higher Tanner scale scores than children without IR. Additionally, children with IR had less favorable profiles of glucose, insulin, and lipoprotein lipids, but a higher TyG index and resting BP than children with IS. Children with IR engage in less VPA than children with IR, with no group difference in LPA or MPA. A bivariate correlation analysis showed that HOMA-IR was significantly and positively correlated with BMI (r = 0.533, *p* < 0.001), WC (r = 0.454, *p* < 0.001), WHtR (r = 0.165, *p* < 0.001), and the TyG index (r = 0.382, *p* < 0.001).

A mediation analysis, as provided in Table 2, revealed that the WHtR coefficient for HOMA-IR was statistically significant. Furthermore, the coefficients of WHtR for the TyG index and the TyG index for HOMA-IR were statistically significant, implying that the TyG index mediates the relationship between the WHtR and HOMA-IR. The bootstrapped 95% CI confirmed that the WHtR had an indirect effect on HOMA-IR through the TyG index (effect = 0.357, SE = 0.079, 95% CI [0.215, 0.528]).

As shown in Table 3, the results of the moderated mediation analysis indicated that the interaction coefficient between the WHtR and gender for the TyG index was statistically significant (β= −1.369, SE = 0.631, 95% CI [−2.608, −0.129]) independent of age and Tanner scale scores. Figure 3 shows that the slope of the TyG index regression on the mean-centered WHtR values was steeper in girls than in boys. This suggests that an increase in WHtR has a greater effect on increasing the TyG index in girls than in boys, even though girls have lower TyG index values relative to the WHtR. When VPA was considered, however, the interaction effect between WHtR and gender for the TyG index was no longer significant (β= −1.197, SE = 0.638, 95% CI [−1.876, 0.0612]).

## 4. Discussion

In this cross-sectional study of 613 Korean children, we examined whether the sex of the children affects the role of the TyG index in determining the relationship between the WHtR and HOMA-IR. Our findings show that the WHtR influences HOMA-IR both directly and indirectly through the TyG index. Notably, our findings show that the extent to which the WHtR influences the TyG index varies by sex: girls are more vulnerable to an increase in the TyG index caused by an increase in the WHtR than boys. Additionally, inequality in VPA may partially contribute to this gender difference in the relationship between the WHtR and the TyG index. Given the cross-sectional nature of this study, however, the gender disparities observed need to be investigated in a cause-and-effect manner.

Similarly to adults, the TyG index has been implicated in various health conditions among children and adolescents in a population-based study of 7404 Korean youths from the 2008–2016 Korean National Health and Nutrition Survey [29]. The TyG index also has diagnostic value in estimating the T2DM risk in 176 overweight and obese youths [30], the severity of hepatic steatosis in youths with a non-alcoholic fatty liver [31], and a metabolically unhealthy phenotype in overweight and obese boys [32].

In favor of the mediating effect of the TyG index observed in our study, the TyG index acted as a mediator in determining the relationship between obesity and IR in a prospective cohort study of 94,136 Chinese adults without CVD aged 18–98 years [33] and a retrospective cohort study of 16,613 US adults aged 18–65 years [34]. Additionally, the mediating effect of the TyG index on the relationship between BMI and incident hypertension was observed in a population-based cohort study of 4081 Chinese adults aged 35 years and older [35] and a 6-year prospective cohort study involving 10,309 Chinese adults aged 18 years [36]. In a 22.7-year population-based cohort study of 176,420 Australian adults aged 18 years and older, the impact of BMI on the incident of end-stage kidney disease was found to be substantially mediated by the TyG index [37].

Concerning gender differences in the impact of the WHtR on the TyG index, some studies have reported sex differences in the relationship between the TyG index and health outcomes. For example, the TyG index was positively associated with the risk of subclinical atherosclerosis and obstructive coronary artery disease in non-diabetic women, but not in non-diabetic men [22,23]. A secondary analysis of 48,230 participants from the China Rui-Ci group showed that while the TyG index was significantly related to the T2D risk in both men and women, the relationship was stronger among women than among men [21]. Collectively, these findings indicate that although the underlying mechanism(s) are unknown, sex may play a role in determining the relationship between obesity and whole-body IR via the TyG index.

Several explanations can be given for this gender difference. First, lifestyle factors, such as how active or inactive a person is, may explain why the TyG index has a different effect on the relationship between obesity and IR in boys and girls. Inequality in physical activity between 7463 boys and 7998 girls from nine countries has previously been reported, and it is driven by moderate-to-vigorous PA [38]. The gender gap in sport and moderate-to-vigorous PA was also reported in 123,809 adults from 28 European countries [25]. Similarly, we also found a gender difference in PA; boys were more physically active at vigorous intensity than girls. Physical inactivity contributes to IR in children through several mechanisms, including overweight and obesity, inflammation, dyslipidemia, oxidative stress, mitochondrial dysfunction, and others [39]. Second, inequalities in the dietary intakes, food preferences, and eating behaviors between boys and girls is another factor to be considered in determining the relationship between upper-body obesity and the TyG index [40,41]. Lastly, our Tanner scale scores show that girls are physically and sexually more mature than boys, with emotional and hormonal changes, as well as physical changes in body composition and breast development. Girls’ sexual maturation may be temporarily associated with growth hormone-mediated increase in fat oxidation and decreased glucose utilization [41], resulting in a higher TyG index in girls than in boys, independent of the amount of body fat or lifestyle factors.

This study has certain limitations. First, the cross-sectional nature of our study precludes any causal explanation of the findings. An interventional study is required to provide a causal explanation of our findings. Second, lifestyle factors may contribute to gender differences in the relationship between obesity and the TyG index. Future studies should consider lifestyle as a confounding factor when determining the gender-specific effect of obesity on the TyG index or IR. Finally, the current findings need to be confirmed for generalizability to a representative sample of children, even though this study had sufficient statistical power to test the research hypotheses.

## 5. Conclusions

In summary, our findings suggest that upper-body obesity influences both direct and indirect whole-body IR via the TyG index in youths. Furthermore, the impact of WHtR on the TyG index varies by sex, implying that girls may be more susceptible to upper-body-obesity-related IR, including a higher TyG index, than boys, perhaps due to inequality in VPA between boys and girls. This sex-based disparity in the effect of obesity on health outcomes should be considered when designing exercise interventions for children. Given the increasing prevalence of childhood obesity and the need for early detection of metabolic risk, the TyG index—being cost-effective and easy to compute—may serve as a practical screening tool in school health programs or community-based pediatric care settings. Future research should investigate its usefulness in longitudinal tracking and intervention planning.

## Figures and Tables

**Figure 1 healthcare-13-00823-f001:**
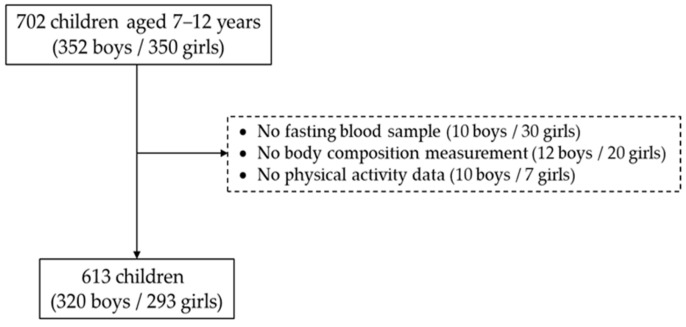
Selection procedure of the study participants.

**Figure 2 healthcare-13-00823-f002:**
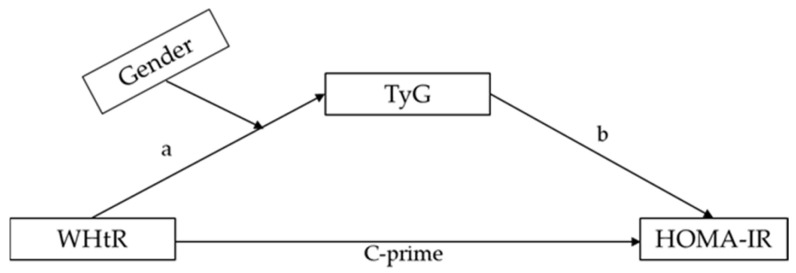
Triglyceride–glucose (TyG) index outperforms body mass index and homeostatic model assessment for insulin resistance (HOMA-IR) in identifying clustered metabolic syndrome (MetS) risk.

**Figure 3 healthcare-13-00823-f003:**
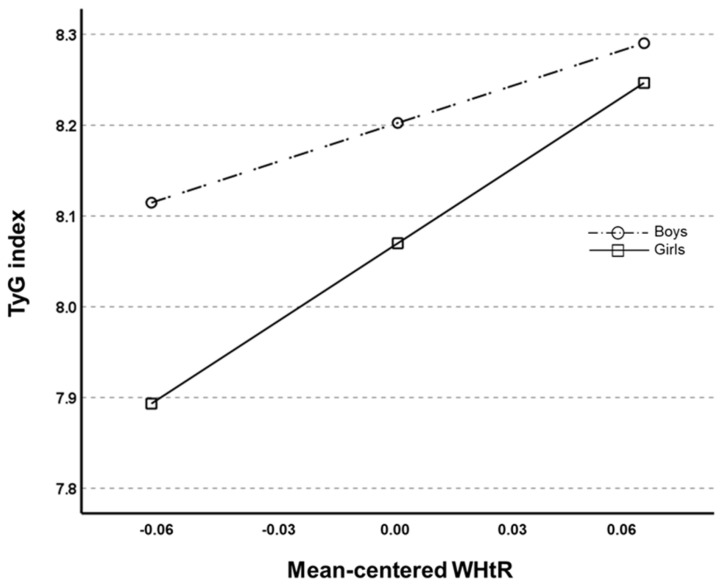
Illustration of the relationship between the triglycerides–glucose index and mean-centered waist-to-hip ratio (WHtR) by gender.

**Table 1 healthcare-13-00823-t001:** Descriptive statistics of insulin-sensitive (IS) and insulin-resistant (IR) children.

Variables	IS (*n* = 533)	IR (*n* = 80)	Total (*n* = 613)	*p* for Group
Age (years)	9.7 ± 1.3	10.7 ± 0.9	9.8 ± 1.3	<0.001
Gender				0.001
Boys, *n* (%)	292 (91.3)	28 (8.8)	320 (52.2)	
Girls, *n* (%)	241 (82.3)	52 (17.7)	293 (47.8)	
BMI (kg/m^2^)	18.7 ± 3.1	22.2 ± 3.7	19.2 ± 3.4	<0.001
WC (cm)	65.7 ± 11.3	76.1 ± 10.3	67.1 ± 11.7	<0.001
WHtR	0.83 ± 0.06	0.85 ± 0.06	0.83 ± 0.06	0.041
Tanner scale scores	1.7 ± 0.9	2.3 ± 1.1	1.8 ± 1.0	<0.001
Systolic BP (mmHg)	108.1 ± 16.0	114.8 ± 16.5	109.0 ± 16.2	0.001
Diastolic BP (mmHg)	65.8 ± 6.6	69.7 ± 13.1	66.3 ± 12.1	0.007
FBG (mg/dL)	90.9 ± 6.6	96.6 ± 7.7	91.7 ± 7.0	<0.001
TC (mg/dL)	172.5 ± 26.7	171.6 ± 32.0	172.4 ± 27.4	<0.001
TGs (mg/dL)	81.2 ± 44.4	106.1 ± 66.2	84.5 ± 48.5	<0.001
HDLC (mg/dL)	59.4 ± 13.2	49.6 ± 10.6	58.1 ± 13.3	<0.001
TyG index	8.10 ± 0.45	8.39 ± 0.52	8.14 ± 0.47	<0.001
Insulin (µU/mL)	7.0 ± 2.9	18.7 ± 6.3	8.5 ± 5.3	<0.001
HOMA-IR	1.6 ± 0.7	4.5 ± 1.5	2.0 ± 1.3	<0.001
Physical activity				
LPA (min/day)	171 ± 41	164 ± 46	170 ± 42	0.151
MPA (min/day)	72 ± 24	76 ± 24	72 ± 24	1.58
VPA (min/day)	33 ± 16	28 ± 18	32 ± 17	0.011

BMI, body mass index; WC, waist circumference; WHtR, waist-to-hip ratio; BP, blood pressure; FBG, fasting blood glucose; TC, total cholesterol; TGs, triglycerides; HOMA-IR, the homeostasis model assessment for insulin resistance; LPA, light physical activity; MPA, moderate physical activity; VPA, vigorous physical activity.

**Table 2 healthcare-13-00823-t002:** Mediation analysis and bootstrap results for indirect effect (*n* = 613).

Variables	Mediation Analysis			
	β	SE	*t*	*p*-Value
WHtR (X) → TyG (M)	1.8154	0.3155	5.7534	<0.001
WHtR (X) → HOMA-IR (Y)	0.7284	0.1480	4.9189	<0.001
TyG (M) → HOMA-IR (Y)	0.1924	0.0197	9.7718	<0.001
Variable	Bootstrap results for indirect effect (X → M → Y)			
	Effect	SE	LL 95% CI	UL 95% CI
TyG	0.3493	0.0748	0.2100	0.5037

X, independent variable; M, mediator variable; Y, dependent variable; WHtR, waist-to-hip ratio; TyG, triglyceride–glucose index; HOMA-IR, the homeostasis assessment model of insulin resistance. The mediation model was adjusted for age, Tanner scale scores, and vigorous physical activity.

**Table 3 healthcare-13-00823-t003:** Moderated mediation analysis and condition indirect effect of WHtR on HOMA-IR by gender (*n* = 613).

Variables	Dependent Variable: TyG			
	B	SE	*t*	*p*-Values
WHtR (X)	4.091	1.003	4.077	0.001
Gender (W)	0.133	0.048	2.784	0.006
Interaction (X × W)	−1.369	0.631	−2.168	0.031

X, independent variable; W, dependent variable; WHtR, waist-to-hip ratio; HOMA-IR, the homeostasis model assessment for insulin resistance. The moderated mediation model was adjusted for age and Tanner scale scores.

## Data Availability

All data used in this study are available from the corresponding author (hkang@skku.edu) upon reasonable request.

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
