# Peer review of "The Role of Gender in the Relationship Between Waist-to-Hip Ratio, Triglyceride–Glucose Index, and Insulin Resistance in Korean Children"

_healthcare, 2025, doi:10.3390/healthcare13070823_

Round 1

Reviewer 1 Report

Comments and Suggestions for Authors

Dear authors, 

I would like to express my gratitude and thank for you for submitting this well designed articles entitled "The Role of Gender on the Relationship between Waist-to-Hip 2, Triglyceride-Glucose Index, and Insulin Resistance in 3 
Korean Children". 

There is missing information that I would like from authors to address it in the manuscript to enhance scientific quality. 

  1. Could do you mention how did you calculate sample size?
  2. Could you mention sampling technique.  snowball or random sampling. 
  3. Could you report response rate and withdrawal percentage. 
  4. Elaborate more based on which physical activity level is classified into light, moderate, and vigorous. 
  5. Could you represent the sociodemographic characteristics of participants' parents.

Author Response

In Our Response to Reviewer # 1

We thank the reviewers for their thoughtful and critical comments. We did our best to address the comments and critics point-by-point, and the revised ones are highlighted in yellow. Four new references have been added and are listed on the last page.

Q1) Could do you mention how did you calculate sample size?

ANS1) Thanks. In our response to the comment, the following statement is added to the Methods:  “The appropriate sample size to test the null hypothesis was calculated as 610 using the G* Power, with effect sizes of.197 (TyG index),.492 (WHtR), and.260 (HOMA-IR), an error probability of.05, and power (1-error probability) of.95.”

Q2) Could you mention sampling technique.  snowball or random sampling. 

ANS2) Thanks. In our response to the comment, the following statement is added to the Methods: 

“This study used convenience sampling to select study participants based on geographical proximity, availability at a given time, or willingness to participate in the study.”

Q3) Could you report response rate and withdrawal percentage.

ANS3) Thanks. In our response to the comment, Response rate was 88%, and dropout rate was 13%.

Q4) Elaborate more based on which physical activity level is classified into light, moderate, and vigorous. 

ANS4) Thanks. In our response to the comment, the following statement is added to the Methods: The uniaxial accelerometer measured PA at nine different levels. Light PA is defined as levels 1 to 3 (estimated METs of 1.8 to 2.9), moderate PA as levels 4 to 6 (estimated METs of 3.6 to 5.2), and vigorous PA as levels 7 to 9 (estimated METs of 6.1 to 8.3).

Q5) Could you represent the sociodemographic characteristics of participants' parents.

ANS5) Thanks. Unfortunately, however, we did not obtain any SES information about the parents.

List of Added References

  1. Tagi, V.M.; Chiarelli, F. Obesity and insulin resistance in children. Curr Opin Pediatr 2020, 32, 582-588.
  2. Newbern, D.; Gumus Balikcioglu, P.; Balikcioglu, M.; Bain, J.; Muehlbauer, M.; Stevens, R.; Ilkayeva, O.; Dolinsky, D.; Armstrong, S.; Irizarry, K.; Freemark, M. Sex differences in biomarkers associated with insulin resistance in obese adolescents: metabolomic profiling and principal components analysis. J Clin Endocrinol Metab 2014, 99, 4730-4739. Erratum in: J Clin Endocrinol Metab 2015, 100, 1709.
  3. Miao, Y.; Wang, Y.; Wan, Q. Association between TyG index with obesity indicators and coronary heart disease: a cohort study. Sci Rep 2025, 15, 8920.
  4. Choi, J.; Yoon, T.W.; Yu, M.H.; Kang, D.R.; Choi, S. Gender and age differences in the prevalence and associated factors of metabolic syndrome among children and adolescents in South Korea. Child Health Nurs Res 2021, 27, 160-170.

Reviewer 2 Report

Comments and Suggestions for Authors

Dear Authors, 

Thank you for demonstrating the usefulness of the Waist-to-Hip Ratio as a simple and practical indicator, particularly in the context of the increasing societal concern regarding childhood obesity. To further enhance the academic and clinical significance of the manuscript, I suggest considering a few revisions.

Please refer to the attached file for detailed comments.

Best regards,

Author Response

In Our Response to Reviewer # 2

We thank the reviewers for their thoughtful and critical comments. We did our best to address the comments and critics point-by-point, and the revised ones are highlighted in yellow. Four new references have been added and are listed on the last page

[Introduction]

The Introduction section presents important concepts such as insulin resistance, HOMA-IR, the TyG index, and sex-related differences. However, the transitions between paragraphs (e.g., from L47 to L48, and from L55 to L56) feel somewhat abrupt. I recommend adding brief bridging sentences to clarify the logical flow between the need for alternative markers of insulin resistance and the relevance of sex differences. This would help readers better understand how each concept builds upon the previous discussion. Additionally, while the background section clearly introduces the theoretical rationale, it does not provide contextual information on the prevalence or trends of childhood obesity in Korea. Including such data—even briefly—could strengthen the rationale for focusing on Korean children and emphasize the potential clinical or public health relevance of using the TyG index as a screening tool

ANS) Thanks. The description of the relevant introduction section is revised and highlighted in yellow, as follows:

Introduction

Childhood obesity is a significant risk factor for insulin resistance (IR), which can lead to type-2 diabetes (T2D) and cardiovascular disease (CVD) later in life [1]. IR is a pathological condition in which the skeletal muscle, adipose tissue, and liver fail to respond normally to insulin [2]. IR is a hallmark of metabolic syndrome (MetS), which is characterized by central obesity, hyperglycemia, hypertension, and dyslipidemia, all of which lead to the development of T2D and CVD [3]. Unhealthy lifestyles, such as excessive caloric consumption, lack of physical activity, and sedentary behavior, have been blamed as the etiological risk factors underlying IR pathology [4].

Several biomarkers, such as fasting insulin, the homeostasis model assessment for IR (HOMA-IR), the quantitative insulin sensitivity check index, and oral glucose tolerance test, have been developed and used to identify IR. Among these, HOMA-IR is the most effective and widely validated biomarker for determining glucose homeostasis within a healthy range [5]. A pathological link between obesity and this biomarker has been reported in pediatric populations [6,7]. Various cut-offs of HOMA-IR for whole-body IR have been developed for Asian children and adolescents, and their clinical utility in identifying youths at a high risk of whole-body IR has been proven in previous studies [8,9]. At the same time, previous research has also found sex differences in the association between the biomarker and obesity-related health conditions [10].

The triglyceride-glucose (TyG) index, a composite measure derived from fasting triglyceride and glucose levels, has been developed and used as an alternative for HOMA-IR [11]. The TyG index has been associated with obesity and related health conditions in adults from the USA [12] and China [13,14]. The TyG index is also positively associated with all-cause and diabetes-specific mortality in people with MetS [15]. The clinical value of the TyG index was critically reviewed and validated in a recent review and meta-analysis of 49,325 participants from 13 observational studies [16]. Compared with the gold standard hyperinsulinemic-euglycemic clamp for measuring IR, the TyG index is also easier to use, takes less time, and is more cost-effective for large-scale use [17]. However, there is little information about how sex influences the relationship between obesity, the TyG index, and whole-body IR in pediatric populations.

Sex-related differences in energy metabolism are well established [18], which may explain sex differences in IR [19,20]. Some studies have indicated that women have a lower TyG index than men, independent of body fat content [21,22], implying that women may be less insulin-resistant or more insulin-sensitive. A high TyG index has been linked to a high risk of subclinical atherosclerosis [23] and obstructive coronary artery disease in women without diabetes [24]. The association between the TyG index and T2D risk is stronger among women than among men [25]. A previous study also reports sex and age differences in the prevalence of obesity and related health conditions in children and adolescents in Korea [26]. Collectively, these findings indicate that sex may influence the relationship between obesity, TyG index, and IR through several factors. This cross-sectional study investigated the influence of sex on the relationship between the waist-to-hip ratio (WHtR), TyG index, and HOMA-IR in a pediatric population using a moderated mediation model.

[Discussion]

The Discussion suggests that gender differences in the impact of WHtR on the TyG index may be explained by inequality in vigorous physical activity (VPA). However, as the study is cross-sectional and VPA was treated as a covariate rather than part of a causal model, this interpretation may be overly deterministic. I recommend softening this causal implication and rephrasing it to suggest that VPA may partially account for the observed gender differences, rather than being a definitive cause. While the moderation analysis initially showed a significant interaction between WHtR and gender on the TyG index, this interaction was no longer significant after adjusting for VPA. This suggests that VPA may be a contributing factor, but the current data do not allow for a definitive causal interpretation. Rewriting this part of the discussion to respect the observational nature of the data and to avoid implying causality would strengthen the manuscript.

ANS) Thanks. The description of the relevant discussion section is softened and highlighted in yellow, as follows:

In this cross-sectional study of 613 Korean children, we examined whether the sex of the children affects the role of the TyG index in determining the relationship between WHtR and HOMA-IR. Our findings show that the WHtR influences HOMA-IR both directly and indirectly through the TyG index. Notably, our findings show that the extent to which the WHtR influences the TyG index varies by sex: girls are more vulnerable to an increase in the TyG index caused by an increase in the WHtR than boys. Additionally, inequality in VPA may partially contribute to this gender difference in the relationship between WHtR and TyG index. Given the cross-sectional nature of this study, however, the gender disparities observed need to be investigated in a cause-and-effect manner.

[Potential Application and Public Health Relevance]

To enhance not only the academic but also the clinical and public health significance of the study, I recommend the authors consider briefly discussing the potential applications of their findings. For instance, the following sentence—or a similar formulation—could be included near the end of the Discussion: “Given the increasing prevalence of childhood obesity and the need for early detection of metabolic risk, the TyG index—being cost-e⒐active and easy to compute—may serve as a practical screening tool in school health programs or community-based pediatric care settings. Future studies should explore its utility in longitudinal tracking and intervention planning.” Such a comment would help connect the study’s findings to real-world practices and broaden its potential impact.

ANS) Thanks. In our response to the comment, we have added the following statements as suggested:

“This sex-based disparity in the effect of obesity on health outcomes should be considered when designing exercise interventions for children. Given the increasing prevalence of childhood obesity and the need for early detection of metabolic risk, the TyG index—being cost-effective and easy to compute—may serve as a practical screening tool in school health programs or community-based pediatric care settings. Future research should investigate its usefulness in longitudinal tracking and intervention planning.”

List of Added References

  1. Tagi, V.M.; Chiarelli, F. Obesity and insulin resistance in children. Curr Opin Pediatr 2020, 32, 582-588.
  2. Newbern, D.; Gumus Balikcioglu, P.; Balikcioglu, M.; Bain, J.; Muehlbauer, M.; Stevens, R.; Ilkayeva, O.; Dolinsky, D.; Armstrong, S.; Irizarry, K.; Freemark, M. Sex differences in biomarkers associated with insulin resistance in obese adolescents: metabolomic profiling and principal components analysis. J Clin Endocrinol Metab 2014, 99, 4730-4739. Erratum in: J Clin Endocrinol Metab 2015, 100, 1709.
  3. Miao, Y.; Wang, Y.; Wan, Q. Association between TyG index with obesity indicators and coronary heart disease: a cohort study. Sci Rep 2025, 15, 8920.
  4. Choi, J.; Yoon, T.W.; Yu, M.H.; Kang, D.R.; Choi, S. Gender and age differences in the prevalence and associated factors of metabolic syndrome among children and adolescents in South Korea. Child Health Nurs Res 2021, 27, 160-170.
